# Maleic Anhydride-Grafted PLA Preparation and Characteristics of Compatibilized PLA/PBSeT Blend Films

**DOI:** 10.3390/ijms23137166

**Published:** 2022-06-28

**Authors:** Hyunho Jang, Sangwoo Kwon, Sun Jong Kim, Su-il Park

**Affiliations:** 1Department of Packaging, Yonsei University, Wonju 26493, Korea; whyhyun@naver.com (H.J.); ksw0089@naver.com (S.K.); 2CJ Cheil Jedang WhiteBio-CJ Research Center, Woburn, MA 01801, USA; ssun0526@gmail.com

**Keywords:** PBSeT, PLA-grafted MAH, compatibilizer, PLA/PBSeT, biodegradable polyester, biopolymer, polymer blends

## Abstract

Poly(butylene sebacate-co-terephthalate) (PBSeT) is a biodegradable flexible polymer suitable for melt blending with other biodegradable polymers. Melt blending with a compatibilizer is a common strategy for increasing miscibility between polymers. In this study, PBSeT polyester was synthesized, and poly(lactic acid) (PLA) was blended with 25 wt% PBSeT by melt processing with 3–6 phr PLA-grafted maleic anhydride (PLA-g-MAH) compatibilizers. PLA-g-MAH enhanced the interfacial adhesion of the PLA/PBSeT blend, and their mechanical and morphological properties confirmed that the miscibility also increased. Adding more than 6 phr of PLA-g-MAH significantly improved the mechanical properties and accelerated the cold crystallization of the PLA/PBSeT blends. Furthermore, the thermal stabilities of the blends with PLA-g-MAH were slightly enhanced. PLA/PBSeT blends with and without PLA-g-MAH were not significantly different after 120 h, whereas all blends showed a more facilitated hydrolytic degradation rate than neat PLA. These findings indicate that PLA-g-MAH effectively improves PLA/PBSeT compatibility and can be applied in the packaging industry.

## 1. Introduction

With growing global environmental problems that are exacerbated by greenhouse gases, bioplastics have received increasing attention as a countermeasure for reducing carbon dioxide emissions from the production and waste of petroleum-based plastics [1,2,3]. Poly(lactic acid) (PLA) can be obtained from fully renewable biomass sources and is one of the most commercially viable representative bio-plastic materials worldwide [4]. PLA has been used in the medical and packaging industries because it combines high mechanical strength, transparency, non-toxicity, and biocompatibility [5,6,7].

However, widespread use of PLA is limited, particularly in the flexible packaging industry, owing to its brittleness, low impact resistance, and slow crystallization rate [8,9,10]. To overcome these drawbacks of PLA, several studies have been conducted to increase ductility via melt blending that employs flexible biodegradable polymers, such as poly(butylene-adipate–co–terephthalate) (PBAT), polycaprolactone (PCL), poly(butylene succinic acid) (PBS), poly(hydroxy alkanoate) (PHA), and poly(butylene-sebacate-co-terephthalate) (PBSeT) [10,11,12,13,14]. Unfortunately, poor interfacial separation occurred in blends of PLA with other biodegradable polymers, resulting in immiscible blends and affecting material mechanical properties and modulus [15,16]. To enhance the miscibility of a blend, one possible strategy involves the use of a compatibilizer.

Hexamethylene diisocyanate (HDI), dioctyl maleate (DOM), maleic anhydride (MAH), multifunctional epoxy, and PLA-grafted maleic anhydride (PLA-g-MAH) have been reported to be effective compatibilizers for PLA-based blends [17,18,19,20,21,22]. PLA-g-MAH has been used as a coupling agent to improve low interfacial adhesion between PLA and PBAT, polypropylene (PP), fiber, rubber, wood flour, and nanocomposite [20,23,24,25,26,27,28,29]. MAH is an appropriate candidate for the synthesis of grafted PLA because it has two functional units capable of esterification reaction and high reactivity toward initiator-induced free radicals [30,31]. It also does not easily undergo self-polymerization under free-radical grafting conditions and exhibits low toxicity [32].

In a previous study, PBSeT block co-polyester synthesized using sebacic acid (Se), dimethyl terephthalate (DMT), and 1,4-butanediol (BDO) showed high flexibility (>800%) [33]. In addition, adding PBSeT with various HDIs was confirmed to enhance the miscibility and the ductility of PLA blend films [14]. However, the presence of HDI in a PLA/PBSeT blend is potentially harmful in food and pharmaceutical packaging applications. Therefore, other compatibilization methods must be developed. In this study, PLA/PBSeT blends were prepared using melt processing in the presence of various amounts of the synthesized PLA-g-MAH using dicumyl peroxide (DCP) as an initiator to improve the miscibility of the blends. Moreover, the mechanical properties, thermal behaviors, and hydrolytic degradation properties of the blends were investigated.

## 2. Results

### 2.1. Fourier Transform Infrared (FT-IR) Spectra of PLA-g-MAH

The Fourier-transform infrared (FT-IR) spectra of neat PLA and PLA-g-MAH are compared in Figure 1, and detailed spectra with various DCP and MAH contents are presented in Figure 2. Characteristic absorption peaks appeared for neat PLA, presented at 3000–2800, 1780–1700, 1500–1320, and 1300–1000 cm^−1^, which correspond to CH, CH_3_ stretching, C = O stretching, CH, CH_3_ bending, and C–O stretching, respectively [30,34]. Meanwhile, PLA-g-MAH exhibited peaks at 1850, 1750, and 695 cm^−1^, and the detailed spectra are shown in Figure 2. The weak intensity of the peak at 1850 cm^−1^ was attributed to the C = O stretching of cyclic anhydride [34,35,36]. As the DCP and MAH contents of PLA-g-MAH increased, the intensity of this peak tended to slightly increase. This absorption peak can be considered a representative peak band in which MAH is grafted onto the PLA backbone [37,38]. Moreover, the increased intensity of the C = O stretching band at 1750 cm^−1^ was attributed to the effect of MAH–PLA bonding [39]. The peak at 695 cm^−1^ corresponding to aromatic CH bending was related to the unreacted anhydride, which could occur at high MAH concentrations in the samples [36]. The intensity of these bands was observed to increase with increasing MAH content for equal DCP contents, whereas the intensity of the peak at 695 cm^−1^ (aromatic CH bending) decreased as the DCP content increased from 0.5 to 1.0 phr. This implies that the DCP concentration is a major factor in the effective synthesis of PLA-g-MAH, as it enhances the grafting yield.

### 2.2. Determination of the Grafting Yield

Table 1 shows the grafting yield of the synthesized PLA-g-MAH samples determined by a titration procedure, along with the grafting efficiency calculated from the actual grafted MAH content and the input amount of MAH. At a constant MAH content, a higher DCP content resulted in an increased grafting yield and efficiency. This implies that the increase in initiator-induced radical formation ensures a higher chain transfer to the PLA backbone, which leads to a higher grafting yield and efficiency [40]. The grafting efficiency tended to decline with increasing MAH content. As verified by FT-IR analysis, more unreacted MAH was generated at high MAH concentrations during the grafting process.

The maximum grafting yield was 0.340% for the PLA-g-MAH composed of 1.0 phr DCP and 6 phr MAH (denoted by DCP1.0-M6; see Table 1 for all sample labels). However, for blends with an MAH content above 6 phr, the grafting yield decreased. Degradation of the grafting yield after the optimum level of MAH content may have resulted from either phase separation between the MAH and the polymer or induced side reactions, such as crosslinking and oligomeric and succinic anhydride end-grafting [38,41,42]. The grafting yield is an important index for estimating the quality of a synthesized grafted polymer when using a compatibilizer. Previous studies have demonstrated that the major factors affecting the grafting process are the initiator and monomer concentrations [36,38,40,43]. In this study, the DCP1.0-M6 sample showed the highest grafting yield and was thus selected as the compatibilizer for the proposed PLA/PBSeT blends. Hereafter, PLA-g-MAH refers to DCP1.0-M6.

### 2.3. Characterization of the PLA/PBSeT/PLA-g-MAH Blends

#### 2.3.1. Mechanical and Morphology Properties

The mechanical properties of neat PLA, PBSeT, and their binary blends were previously reported [14,33]. In the case of simple blending without a compatibilizer, the mechanical properties of soft biodegradable polymers are not fully exploited in the blends because of the low miscibility of PLA and other polymers [26,30,34]. In this work, to overcome these problems, various contents of synthesized PLA-g-MAH were applied as a compatibilizer to PLA/PBSeT blends.

Figure 3 shows the mechanical properties of neat PLA, PBSeT, a PLA/PBSeT blend film without PLA-g-MAH (hereafter designated as the control), and PLA/PBSeT blend films with various PLA-g-MAH contents. All blend films presented higher tensile strengths than PBSeT alone and greater elongation at break than neat PLA. The tensile strengths of neat PLA, PBSeT, and the control film were about 52.6, 15.9, and 23.8 N/mm^2^, respectively. Meanwhile, the PLA-g-MAH-compatibilized PLA/PBSeT blend films exhibited increased tensile strength. There was a statistically significant difference (*p* < 0.5) between all samples according to a one-way analysis of variance (ANOVA) with a post hoc Scheffe’s test.

The elongation at break of PLA/PBSeT blend films with PLA-g-MAH showed a gradual increase with an increasing PLA-g-MAH content; the average elongation at break of PLA; PBSeT; control; and the PLA/PBSeT blends with PLA-g-MAH 3, 6, and 9 phr were approximately 2.4%, 868.3%, 45.9%, 59.3%, 75.1%, and 119.3%, respectively. Although the elongation characteristics of PLA/PBSeT blend films were improved by the addition of PLA-g-MAH, there was no statistically significant difference below 3 phr PLA-g-MAH sample and the control.

This mechanical behavior might have occurred due to the ester bond reaction between the carboxyl group, which resulted from the ring opening of anhydride in PLA-g-MAH, and the hydroxyl groups of PLA and PBSeT [34]. Moreover, the unreacted MAH molecule of PLA-g-MAH could have acted as a plasticizer for the blends, affecting the elongation at break [44]. According to these mechanisms, adding the appropriate level of PLA-g-MAH led to an improvement of the mechanical and morphological properties of the blends.

The morphological characteristics of the control and the PLA/PBSeT and PLA/PBSeT/PLA-g-MAH blends with different PLA-g-MAH contents were evaluated by scanning electron microscopy (SEM), as shown in Figure 4. The SEM images of the fracture surfaces of the samples showed many isolated PBSeT droplets in the PLA matrix. This feature indicates phase separation and poor interfacial interaction between the PBSeT domain and PLA matrix. However, the droplet size in the compatibilized PLA/PBSeT blends gradually decreased with an increasing PLA-g-MAH content. Moreover, it was shown that the dispersion and distribution of the PBSeT domain in the PLA matrix appeared more evenly with an increased amount of added PLA-g-MAH. The reduction in droplet size reflected the enhanced interfacial adhesion between the polymers [45]. Interfacial adhesion could be improved through the trans-esterification reaction within the hydroxyl groups of PLA and PBSeT and the anhydride group of PLA-g-MAH [30]. The well-dispersed domain and increased interfacial adhesion thus promoted the tensile strength and toughness of the blends, which supported the results obtained for the mechanical properties of PLA/PBSeT blend films with PLA-g-MAH. In other words, these results indicate the effectiveness of PLA-g-MAH as a compatibilizer for the PLA/PBSeT blend.

#### 2.3.2. Differential Scanning Calorimetry (DSC)

The thermal properties of PLA/PBSeT/PLA-g-MAH blends were also determined by differential scanning calorimetry (DSC). Figure 5 presents a DSC thermogram from the second heating run curve, as the first run was used to eliminate thermal history, and the detailed results are shown in Table 2. The PLA/PBSeT blends with PLA-g-MAH show multiple melting temperature (T_m_) peaks, with the first melting peak (T_m1_) at approximately 31 °C attributed to the PBSeT component and the second and third melting peaks (T_m2_ and T_m3_) corresponding to the PLA-g-MAH and PLA components. Similar thermal behaviors have been observed in several studies [20,30,34]. The phenomenon of two overlapped T_m_ peaks in the case of PLA-g-MAH suggests that the grafted part of PLA affected the formation of PLA crystals, which might be explained by a melt and re-crystallization mechanism [46]. Moreover, crystallization could be affected by molecular weight, cooling temperature, and crystallization temperature [20]. Some authors have reported a reduction in the molecular weight during the grafting process [47,48]. Evidently, PLA-g-MAH presented a lower T_m_ value than neat PLA, along with double melting peaks. This is because PLA-g-MAH contained different degrees of crystallinity owing to the presence of the grafted component, which resulted in lower mobility than that of neat PLA [49].

The glass transition temperatures (T_g_) of neat PLA, PLA/PBSeT, and PLA/PBSeT/PLA-g-MAH blends were 62.9, 60.1, 61.2, 60.4, and 59.2 °C, respectively. The reduced T_g_ of the PLA/PBSeT blends with and without PLA-g-MAH resulted in an increased elongation at break.

The degree of crystallinity (X_c_) and enthalpy of cold crystallization increased with PLA-g-MAH content. This improved crystallization resulting from the addition of PLA-g-MAH as a compatibilizer for PLA-based blends has been reported in other studies [30,50]. It was speculated that PLA-g-MAH might act as a nucleating agent, thus providing stronger and easier crystallization than those in a simple PLA/PBSeT blend.

#### 2.3.3. Thermal Gravimetric Analysis (TGA)

The thermal stabilities of neat PLA, PBSeT, and the PLA/PBSeT blends with and without PLA-g-MAH were determined by thermogravimetric analysis (TGA). Figure 6 presents the TGA result and derivative thermogravimetry (DTG) thermogram. Moreover, Table 3 lists the onset decomposition temperature (T_onset_); thermal decomposition temperatures at weight loss percentages of 25%, 50%, and 75% (T25, T50, and T75); and the temperatures at which the maximum thermal decomposition rate occurred (T_max_). PLA and PBSeT underwent a single-step decomposition, while the PLA/PBSeT (control) and PLA/PBSeT/PLA-g-MAH blends exhibited two decomposition steps owing to the presence of the PLA and PBSeT components in the blends. The higher T_max_ of PBSeT was caused by the aromatic phthalate in its molecular structure. Furthermore, the T_max_ of neat PLA and PBSeT were approximately 340 and 400 °C, as also observed in previous research [33].

The T_onset_ of the simple PLA/PBSeT blend (control) and the PLA/PBSeT blend with PLA-g-MAH 3, 6, and 9 phr were 334.6, 339.2, 337.4, and 337.7 °C, respectively; thus, the addition of PLA-g-MAH increased T_onset_. A similar tendency was observed for 25% weight loss. Together, these results indicate that thermal stability was slightly improved by the presence of PLA-g-MAH in the blends.

### 2.4. Hydrolytic Degradation Test

Hydrolysis testing is used to indirectly predict the biodegradability and biodegradation rate of biodegradable polyesters [51]. The hydrolytic degradation of such polyesters as a part of abiotic degradation occurs through the hydrolysis of ester bonds in water [52]. Several studies have performed hydrolytic degradation tests under acid and basic conditions to accelerate the degradation rate [53,54]. In this study, an accelerated hydrolysis test was conducted in an aqueous solution of 0.1 N NaOH (pH 13) at 37 °C. Figure 7 and Figure 8 present the hydrolytic degradation curves and appearances, respectively, of neat PLA and PLA/PBSeT blend films during hydrolytic degradation. The PLA/PBSeT blend films mostly degraded within days, and neat PLA exhibited considerably lower weight loss than the PLA/PBSeT blends throughout the test period.

Hydrolysis can occur by surface and bulk erosion, which depend on the diffusion rate of water molecules [55]. Bulk erosion occurs when the water absorption of the specimens is faster than the degradation of its polymer chains, which breaks the specimen into smaller pieces [56,57]. As shown in Figure 8, the disintegrated specimens and the holes in the films indicate that all samples undergo both bulk and surface erosion. The hydrolytic degradation rate is also related to polymer crystallinity. Fundamentally, hydrolytic degradation is known to preferentially occur in amorphous regions rather than crystalline regions because the hydrolysis medium has more access to polymer chains in the less-ordered structure.

From a morphological perspective, phase separation in incompatibilized polymer blends leads to void formation; thus, water can easily penetrate the polymer matrix, resulting in a higher rate of hydrolysis [58,59]. The hydrolytic degradation of the PLA/PBSeT blend with PLA-g-MAH was expected to be delayed relative to that of the PLA/PBSeT blend because of the reduced interfacial space caused by the enhanced interfacial interaction and increased crystallinity, which was confirmed by morphological and thermal property characterization. However, there was no significant difference in the hydrolytic degradation rate between the PLA/PBSeT blends with and without PLA-g-MAHs after 48 h. These results indicate that the increased interfacial adhesion and crystallinity due to the addition of PLA-g-MAH did not sufficiently delay water diffusion within the polymer matrix. In addition, the bulk erosion might have accelerated during the test period because of the thinness of the film specimen. These factors may account for the lack of clear differentiation in the hydrolytic degradation rate among the blends. Future research must clarify the effect of PLA-g-MAH on the degradation rate through hydrolytic degradation tests applied to sheet specimens and biodegradation tests in a real environment involving soil and compost conditions.

## 3. Materials and Methods

### 3.1. Materials

Bio-based Se was provided by Daejung Chemical & Metals Co., Ltd. (Siheung, Korea). DMT was obtained from SK Chemicals (Seoul, Korea). Titanium tetrabutoxide (TBT) as a catalyst for PBSeT synthesis, DCP as a radical initiator of the grafting reaction, and MAH were obtained from Sigma-Aldrich (Saint Louis, MO, USA). PLA was obtained from NatureWorks LLC (PLA 4032D grade, Minnetonka, MN, USA).

### 3.2. Synthesis of PBSeT

PBSeT was synthesized through two-step esterification and subsequent polycondensation under high-vacuum conditions using a 1 L three-necked reactor according to the method reported by Kim et al., with a slight modification [33]. First, 1.25 mol% BDO and 0.4 mol% DMT were esterified with 0.37 g/mole TBT as the catalyst at 180–200 °C. After the first esterification, 0.6 mol% sebacic acid and 0.37 g/mole TBT was added, and the mixture was heated to 200–220 °C. The esterification was terminated after obtaining a theoretically determined quantity of the byproduct. Polycondensation was then initiated in the controlled vacuum state by increasing the temperature to 270 °C. The molar ratio of BDO/dicarboxylic acid was set to 1.25:1. The synthesizing strategies and predicted structure are illustrated in Figure 9.

### 3.3. Preparation of MAH-Grafted PLA (PLA-g-MAH)

The MAH was grafted onto PLA at 190 °C for 12 min using an internal batch mixer (TEST ONE, Siheung, Korea) at a screw speed of 55 rpm. PLA was initially introduced into the mixing chamber, followed by the addition of the DCP as an initiator to generate free radical sites. After allowing free radical generation to proceed for 2 min, the MAH was added as a monomer to the mixture to graft the MAH onto the PLA backbone. The experimental formulations for PLA-g-MAH are shown in Table 4.

### 3.4. Fourier Transform Infrared (FT-IR) Spectroscopy of PLA-g-MAH

FT-IR spectra were recorded at room temperature using a Spectrum 65 FT-IR spectrometer (PerkinElmer, Inc., Waltham, MA, USA). The spectra for the PLA-g-MAH were measured in the range of 4000 to 400 cm^−1^ with 32 scans and a resolution of 2 cm^−1^.

### 3.5. Determination of Grafting Yield

The grafting yield was determined by a titration method related to the acid number. PLA-g-MAH samples (1 g) were completely dissolved in 100 mL of chloroform and 1 mL of 1 N hydrochloric acid in water. The solution was vigorously stirred for 30 min. The grafted PLA samples were purified by precipitation in 500 mL of methanol to remove residual or unreacted MAH and DCP. The precipitates were collected by vacuum filtration using a glass microfiber filter and then dried in a vacuum oven at 80 °C for 12 h. For each purified sample, 1 g was dissolved in a mixture of chloroform and methanol (5:2, *v*/*v*), and eight drops of a 1% phenolphthalein solution in ethanol were added as an indicator. The solutions were titrated using a 0.1 N potassium hydroxide (KOH) standard solution in methanol. The grafting yield was calculated according to Equations (1) and (2) [38,60]:Acid number (mg KOH/g) = V_KOH_ × 56.1/g Sample(1)
Grafting yield = (Acid number × 98.06)/(2 × 561)(2)
where V_KOH_ is the volume (mL) of the 0.1 N KOH methanolic standard solution.

### 3.6. Preparation of PLA/PBSeT/PLA-g-MAH Blend Films

The PLA/PBSeT/PLA-g-MAH blend films were prepared using the internal batch mixer at 80 rpm and 190 °C for 5 min. Before blending, all components were dried overnight at 60 °C under vacuum. The possible reaction mechanisms of partial blends illustrated in Figure 10 and Table 5 show the blend compositions. Specimens employed to characterize the blends were hot-pressed at 195 °C and 25 MPa for 3 min, and specimens with a thickness of 180–200 μm were selected for further analysis.

### 3.7. Characterization of PLA/PBSeT/PLA-g-MAH Blends

#### 3.7.1. Mechanical and Morphology Properties

The mechanical properties of specimens at room temperature were analyzed using a universal testing machine (Qumesys, Anyang, Korea) with a crosshead speed of 50 mm/min. A dumbbell-shaped specimen was employed according to the ISO 527 standards. At least five specimens were tested for each sample, and the mean and standard deviation were calculated.

The cross-section morphologies of the blend specimens were examined using SEM (TM3000, Hitachi, Tokyo, Japan). All samples were mounted onto a cylindrical aluminum probe using conductive double-sided carbon tape and observed at an accelerating voltage of 15 kV.

#### 3.7.2. DSC

DSC thermograms were measured using a DSC-Q20 calorimeter (TA Instruments, Milford, MA, USA) under a nitrogen atmosphere to determine the thermal transition and crystallization behaviors. Samples (5 mg) were sealed in an aluminum pan and then heated and cooled at 10 °C/min within a temperature range of −50 to 210 °C. The melting temperature (T_m_), glass transition temperature (T_g_), and cold crystallization temperature (T_cc_) were determined by the second heating scan. The crystallinities (X_c_) of the samples were evaluated using Equation (3):X_c_(%) = ΔH_m_/(0.25 ΔH^0^_m1_ + 0.75 ΔH^0^_m2_) × 100(3)
where ΔH_m_ is the melting enthalpy of the samples from the first heating scan, and ΔH^0^_m1_ and ΔH^0^_m2_ are the enthalpies of fully crystalline PLA (93.7 J/g) and PBSeT (144.4 J/g), respectively [61,62].

#### 3.7.3. TGA

TGA was performed using a TGA 4000 thermogravimetric analyzer (PerkinElmer, Waltham, MA, USA) to determine the thermal stability of the samples. Each specimen (10 mg) was heated in the range from 30 to 800 °C at a heating rate of 10 °C/min under a nitrogen atmosphere.

### 3.8. Hydrolytic Degradation Test

Accelerated hydrolytic degradation tests were performed in a 0.1 N sodium hydroxide (NaOH, pH = 13) solution at 37 ± 0.2 °C following the method reported by Wang et al. [63]. The specimen length, width, and thickness were 150, 25, and 0.2 mm, respectively. All the samples were weighed before degradation testing and immersed in the 0.1 N NaOH solution for five days, with stirring. Every 24 h, samples were removed from the solution, washed in fresh water, carefully wiped with a paper towel, dried in a vacuum chamber at 60 °C to remove residual moisture, and reweighed. The degree of hydrolytic degradation was expressed in terms of the weight loss using Equation (4):Weight loss (%) = [(*W*_0_ − *W*_t_)/*W*_0_] × 100(4)
where *W*_0_ is the initial weight of the sample, and *W*_t_ is the residual weight at time t. All specimens were measured in triplicate, and the data closest to the mean degradation were reported.

## 4. Conclusions

A PLA-g-MAH blend material was successfully synthesized using an internal batch mixer. The optimum composition of PLA-g-MAH contained 0.1 phr DCP and 6 phr MAH, which provided the highest grafting yield (0.340%) among those obtained under the designated test conditions. Adding over 6 phr PLA-g-MAH into the PLA/PBSeT blend significantly enhanced the tensile strength and elongation at break. In particular, 9 phr g-MAH in a PLA/PBSeT blend provided the highest elongation (119.3%). This is approximately 50 times higher than that of neat PLA and 2.5 times higher than that of the simple PLA/PBSeT blend with no compatibilizer. The morphologies of the compatibilized PLA/PBSeT blends indicated that the PBSeT domain sizes were reduced and better dispersed in the PLA matrix as a result of improved interfacial adherence. In addition, the PLA-g-MAH-compatibilized PLA/PBSeT blends showed increased crystallinity (X_c_) and thermal stability. The PLA/PBSeT blends with and without PLA-g-MAH showed more facilitated hydrolytic degradation than neat PLA, and compatibilization with PLA-g-MAH did not affect the hydrolytic degradation rates of the blends. Consequently, these results suggest that PLA-g-MAH is a promising compatibilizer for PLA/PBSeT blends and that the proposed compatibilized PLA/PBSeT blend films may be suitable for use as packaging materials.

## Figures and Tables

**Figure 1 ijms-23-07166-f001:**
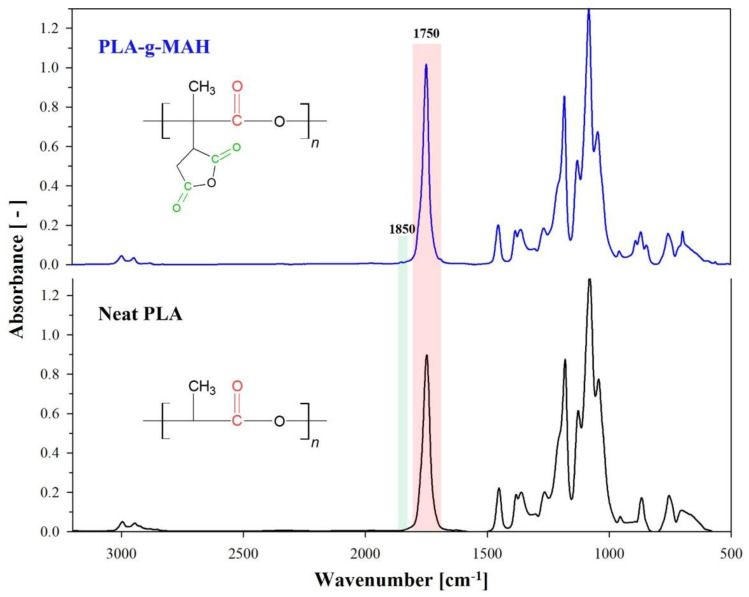
FT-IR spectrum of neat PLA and PLA-g-MAH with 1 phr DCP and 6 phr MAH.

**Figure 2 ijms-23-07166-f002:**
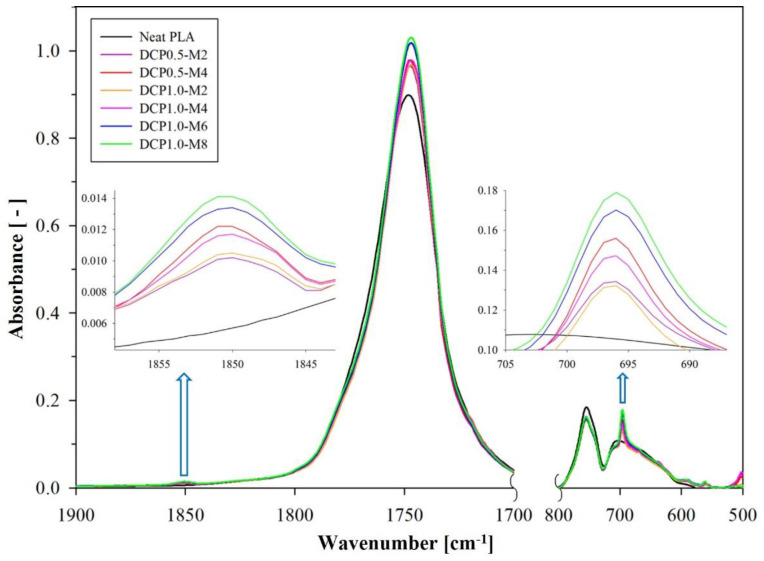
Details of FT-IR spectrum (1900–1700 cm^−1^ and 800–500 cm^−1^) for neat PLA and PLA-g-MAH with variable MAH and DCP contents.

**Figure 3 ijms-23-07166-f003:**
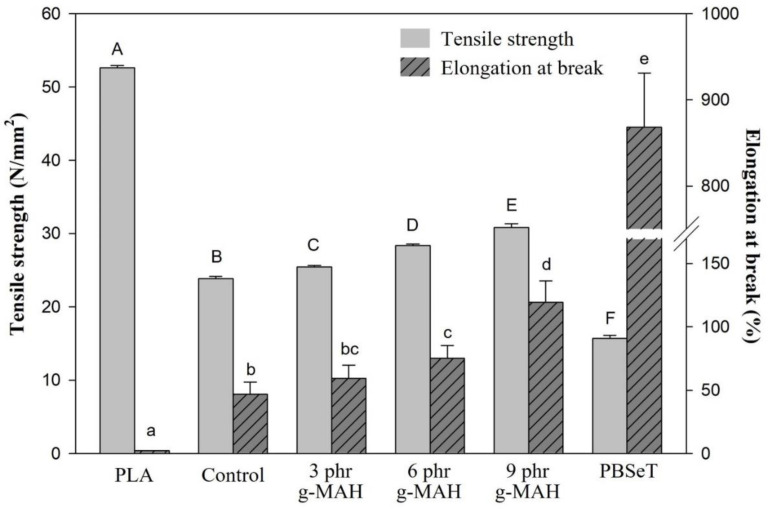
Mechanical properties of PLA/PBSeT and PLA/PBSeT blend films without PLA-g-MAH (control) and with different levels of PLA-g-MAH contents. Uppercase letters and lowercase letters represent significant differences between tensile strength and elongation at break, respectively (*p* < 0.5).

**Figure 4 ijms-23-07166-f004:**
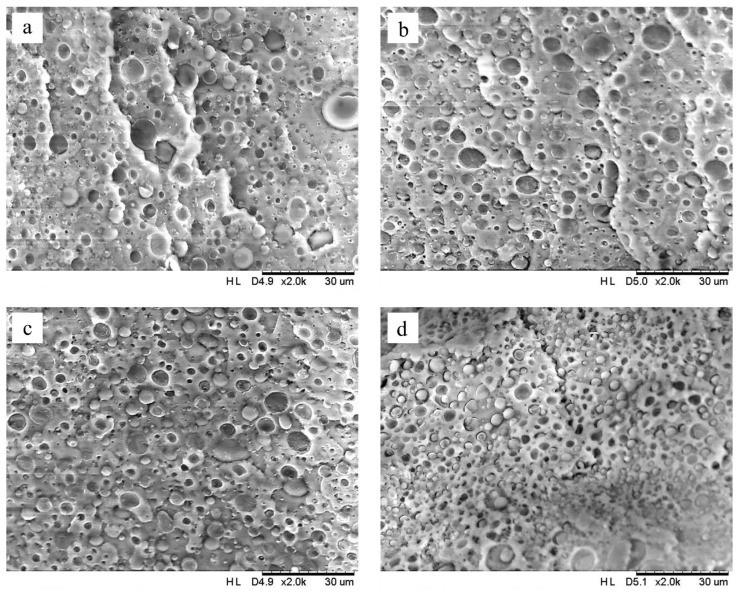
SEM image at 2000× magnification of the fracture surfaces: (**a**) the control PLA/PBSeT blend film; PLA/PBSeT blend films with (**b**) 3 phr g-MAH, (**c**) 6 phr g-MAH, and (**d**) 9 phr g-MAH.

**Figure 5 ijms-23-07166-f005:**
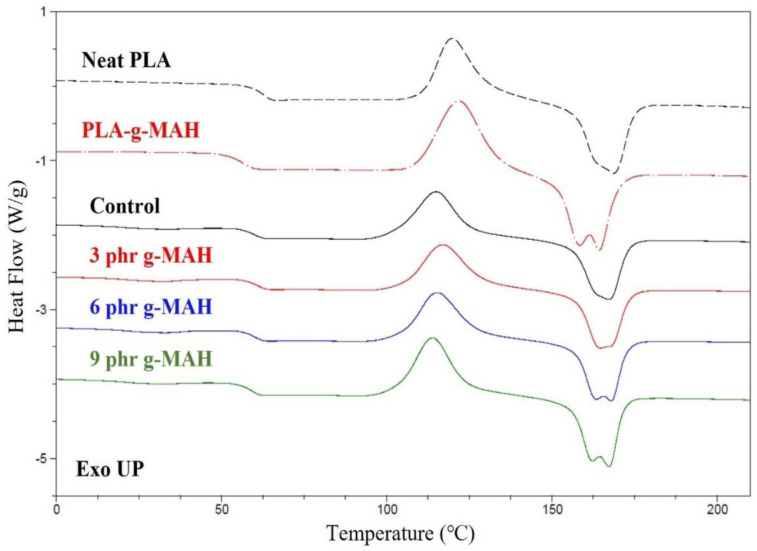
DSC thermogram of neat PLA, PLA-g-MAH, and PLA/PBSeT blend films without PLA-g-MAH (control) and with various PLA-g-MAH contents.

**Figure 6 ijms-23-07166-f006:**
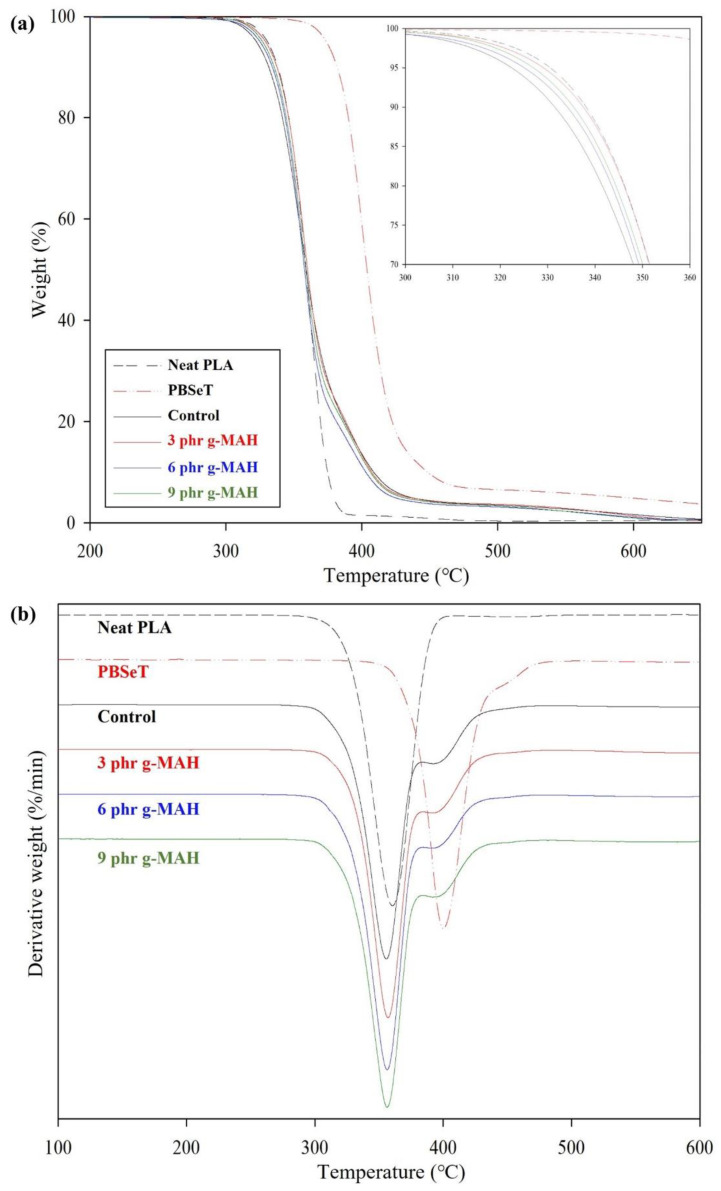
(**a**) TGA and (**b**) DTG thermograms of neat PLA, neat PBSeT, and PLA/PBSeT blends without PLA-g-MAH (control) and with various PLA-g-MAH contents.

**Figure 7 ijms-23-07166-f007:**
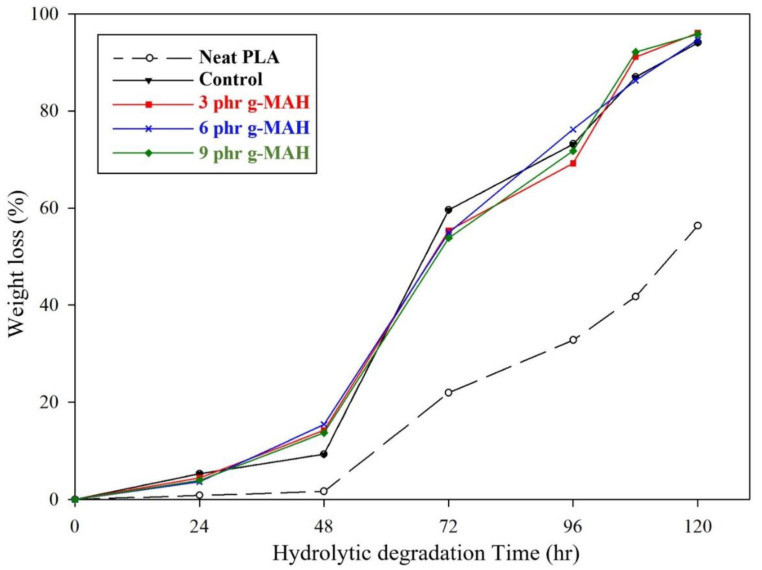
Hydrolytic degradation curves of neat PLA, PBSeT, and the PLA/PBSeT blend films with and without PLA-g-MAH.

**Figure 8 ijms-23-07166-f008:**
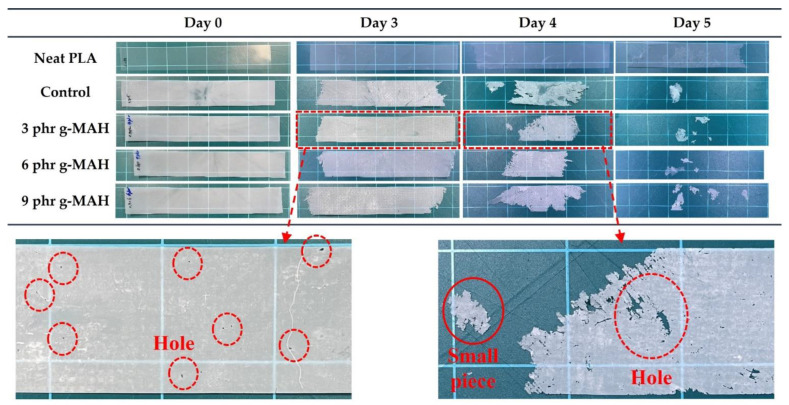
Visual appearances of the neat PLA, PBSeT, and PLA/PBSeT blend films without PLA-g-MAH (control) and with PLA-g-MAH during hydrolytic degradation in 0.1 N NaOH aqueous solution.

**Figure 9 ijms-23-07166-f009:**
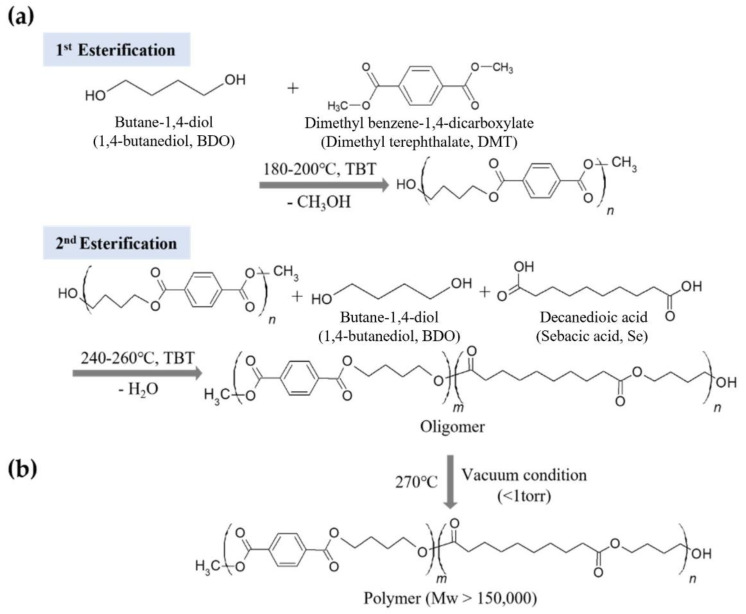
Procedure for the synthesis of PBSeT. (**a**) Step 1: esterification, and (**b**) step 2: polycondensation.

**Figure 10 ijms-23-07166-f010:**
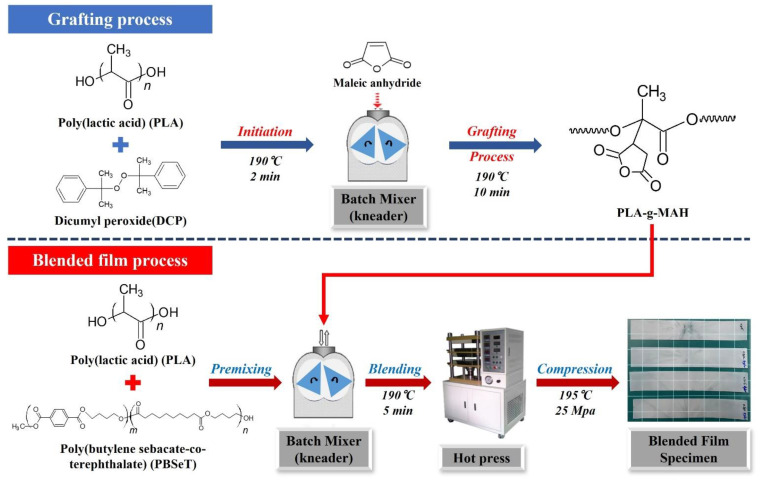
Experimental processes for (**top**) PLA grafting with DCP and MAH and (**bottom**) PLA/PBSeT/PLA-g-MAH blend formation.

**Table 1 ijms-23-07166-t001:** Grafting yield of PLA-g-MAH samples.

Sample	Variable		Acid Number (mg KOH/g)	Grafting Yield (%)	Efficiency (%)
	C_DCP_ (phr)	C_MAH_ (phr)			
DCP0.5-M2	0.5	2	1.91 ± 0.01	0.167 ± 0.001	8.6
DCP0.5-M4	0.5	4	2.08 ± 0.07	0.179 ± 0.006	4.7
DCP1.0-M2	1.0	2	2.86 ± 0.01	0.250 ± 0.001	12.9
DCP1.0-M4	1.0	4	3.53 ± 0.02	0.309 ± 0.002	8.1
DCP1.0-M6	1.0	6	3.89 ± 0.03	0.340 ± 0.003	6.1
DCP1.0-M8	1.0	8	3.09 ± 0.04	0.270 ± 0.004	3.7

**Table 2 ijms-23-07166-t002:** Thermal characteristics from the DSC analysis of PLA/PBSeT blends without PLA-g-MAH (control) and with various PLA-g-MAH contents.

Sample	Thermal Transitions (°C)					ΔH_cc_ (J/g)	ΔHm (J/g)	X_c_ (%) ^a^
	T_g_	T_cc_	T_m1_	T_m2_	T_m3_			
Neat PLA	62.9	119.8	-	-	168.8	34.1	34.8	36.6
PLA-g-MAH ^b^	56.6	121.7	-	158.1	164.3	34.4	35.3	34.7
Control	60.1	114.9	31.6	-	167.2	29.5	30.4	25.4
3 phr g-MAH	61.2	117.0	32.3	164.7	168.0	29.5	30.6	26.8
6 phr g-MAH	60.4	115.3	32.3	163.5	167.9	31.1	31.5	27.5
9 phr g-MAH	59.2	113.9	31.0	162.3	167.2	33.5	34.2	30.3

^a^ Crystallinity was calculated on the basis of the melting enthalpy of the first heating run. ^b^ DCP1.0-M6 showed the highest grafting yield among the PLA-g-MAH samples and was thus used to fabricate films for further analysis.

**Table 3 ijms-23-07166-t003:** TGA data of PLA/PBSeT blends without PLA-g-MAH (control) and various PLA-g-MAH contents.

Sample	T_onset_ (°C)	T_25_ ^a^ (°C)	T_50_ (°C)	T_75_ (°C)	T_max_ (°C)	
					Step 1	Step 2
Neat PLA	339.2	347.6	358.6	368.6	360.5	-
PBSeT	383.1	392.9	403.8	417.4	400.2	-
Control	334.6	345.0	359.0	379.0	355.7	395.9
3 phr g-MAH	339.2	349.0	360.1	379.0	357.0	393.7
6 phr g-MAH	337.4	346.5	357.8	372.6	356.5	394.9
9 phr g-MAH	337.7	347.5	358.7	376.2	356.4	397.0

^a^ The temperature at a given weight loss represented by the subscript (25%, 50%, or 75%).

**Table 4 ijms-23-07166-t004:** Formulations of PLA-g-MAH samples.

DCP (phr)	MAH (phr)			
	2	4	6	8
0.5	DCP0.5-M2	DCP0.5-M4	-	-
1	DCP1.0-M2	DCP1.0-M4	DCP1.0-M6	DCP1.0-M8

**Table 5 ijms-23-07166-t005:** Compositions of PLA/PBSeT/PLA-g-MAH blends.

Sample	PLA% (w/w)	PBSeT% (w/w)	PLA-g-MAH (phr)
Control	75	25	0
3 phr g-MAH	75	25	3
6 phr g-MAH	75	25	6
9 phr g-MAH	75	25	9

## Data Availability

Not applicable.

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
