# Peer review of "Maleic Anhydride-Grafted PLA Preparation and Characteristics of Compatibilized PLA/PBSeT Blend Films"

_ijms, 2022, doi:10.3390/ijms23137166_

Round 1
Reviewer 1 Report
The manuscript is interesting and relates to the miscibility of the PLA/PBSeT polymer composition and its properties.
I have some comments:
1. Line 40-42. Please check this sentence.
2. Line 52, Line 243, Line 113: “we had…”, “We expected…”, “in our previous studies” - The use of phrases such as "we" and "our" is rather not preferred in scientific publications. Please correct.
3. In Introduction, please describe the scientific novelty of research.
4. Figure 1. Please check that the y-axis signature is not missing. Why in the figure 1 the FTIR spectra is in transmittance and in figure 2 in absorbance?
5. Analysis of the crystallinity of the samples with other methods such as WAXS (wide angle X-ray scattering) and SAXS (small angle X-ray scattering) could increase the value of the manuscript.
Author Response
1. Line 40-42. Please check this sentence.
(Answer) Thank you for this helpful comment. We have edited this the sentence on lines 42-43 according to your suggestion.
2. Line 52, Line 243, Line 113: “we had…”, “We expected…”, “in our previous studies” - The use of phrases such as "we" and "our" is rather not preferred in scientific publications. Please correct.
(Answer) In accordance with your suggestion, we revised sentences on lines 53-54, 114-115, and 247-248 to avoid using the first-person plural voice.
3. In Introduction, please describe the scientific novelty of research.
(Answer) Thank you for your comment. We added the following text to the manuscript to describe the novelty of this study on lines 55-58.: “In addition, adding PBSeT with various HDIs was confirmed to enhance the miscibility and the ductility of PLA blend films [14]. However, the presence of HDI in PLA/PBSeT blend is potentially harmful for food and pharmaceutical packaging applications. Therefore, other compatibilization methods must be developed.”
4. Figure 1. Please check that the y-axis signature is not missing. Why in the figure 1 the FTIR spectra is in transmittance and in figure 2 in absorbance?
(Answer) Thank you for your careful review. Accordingly, we have changed the FTIR spectra (Figure 1) such that it is in terms of transmittance instead of absorbance.
5. Analysis of the crystallinity of the samples with other methods such as WAXS (wide angle X-ray scattering) and SAXS (small angle X-ray scattering) could increase the value of the manuscript.
(Answer) We greatly appreciate your suggestion of analyzing our samples using WAXS and SAXS. We understand that these are very effective methods for characterizing the crystallinity of polymers. However, it is very difficult to conduct WAXS and SAX analyses during this limited revision period owing to the lack of accessibility to the WAXS and SAXS equipment and the busy schedule of the analysis center facility. We will employ WAXS and SAXS in our future composite studies. Please understand our situation and appreciate your considerations.
Reviewer 2 Report
suggestions - Please see the attached
It is important that the "Experiment section" is before the "Results section"
Please improve the quality of the figures - Please refer to ref. 21 for similar

Author Response
1. Delete highlight section
(Answer) We deleted this section as you suggested.
2. Rephrase the highlighted sentence and move here- "PBSeT with the compatibilizer HDI showed poor miscibility in PLA"
(Answer) Thank you for your helpful advice. We have rephrased this sentence (lines 55-58). HDI acts as a good potential compatibilizer but may be harmful to human health. We thus attempted to find an alternative compatibilizer in this study. We have added the following text to clarify the aim of our study: “In addition, adding PBSeT with various HDIs was confirmed to enhance the miscibility and the ductility of PLA blend films [14]. However, the presence of HDI in PLA/PBSeT blend is potentially harmful for food and pharmaceutical packaging applications. Therefore, other compatibilization methods must be developed.”
3. Mentioning about Dicumyl peroxide (DCP) used as the initiator of grafting reaction is important here.
(Answer) We mentioned the DCP according to your suggestion (line 58-62): “In this study, PLA/PBSeT blends were prepared using melt processing in the presence of various amounts of the synthesized PLA-g-MAH using dicumyl peroxide (DCP) as an initiator to improve the miscibility of the blends. Moreover, the mechanical properties, thermal behaviors, and hydrolytic degradation properties of the blends were investigated.”
4. Experimental - Materials and Methods should be before the Results section. pre drying conditions of the materials
(Answer) According to the journal guidelines, the Material and Methods section should be placed after the Results and Discussion sections (please see the screenshot below); therefore, we have retained this section in its original location. However, we have added the pre-drying conditions on lines 323-324: “Before blending, all components were dried overnight at 60 °C under vacuum.”
Ref. https://www.mdpi.com/journal/ijms/instructions#preparation
5. DCP -Dicumyl Peroxide.
(Answer) Thank you for your comment, we have added this definition as per your comment (line 60).
6. Suggest graphs with better high resolution -e.g., see ref. 21 figures.
(Answer) Thank you for your kind suggestion. We have replaced Figure 2 with a higher-resolution one, as you recommended. Because Figure 2 was the only graph that was of low resolution
7. better definition to DP10M4 is needed - DP10 - is this DCP -1phr? with 6 phr of MAH -low efficiency - is this due to excess?
(Answer) We replaced labels such as DP10M4 with DCP1.0-M4, and have more clearly defiened our sample labels in the manuscript (see line 99, 107-109, Table 1, Figure 2 and Table 4): “The maximum grafting yield was 0.340% for the PLA-g-MAH composed of 1.0 phr DCP and 6 phr MAH (denoted by DCP1.0-M6; see Table 1 for all sample labels)”.
The lower grafting efficiency of DCP1.0-M6 than that of DCP1.0-M4 is discussed on lines 95-97. As you suggested, it may be due to the excess MAH: “The grafting efficiency tended to decline with increasing MAH content. As versified by FT-IR analysis, more unreacted MAH was generated at high MAH concentrations during the grafting process.”
8. no mention about 9phr in Table 1 -should 9phr be corrected in the Table ? Table 1 shows best grafting efficiency (GE) is with 6phr MAH. With low % of GE with increasing % of MAH here in Fig. 3 the increase in mechanical properties contradicts the conclusions
(Answer) To synthesize PLA-g-MAH, up to 8 phr of MAH was used. In addition, to fabricate PLA/PBSeT/PLA-g-MAH, up to 9 phr of PLA-g-MAH was added during the blending process in this study.
The grafting yield was considered when selecting the optimal amount of PLA-g-MAH. DCP1.0-M6 showed the highest grafting yield and was therefore used to fabricate films for further analysis, as mentioned on lines 106-109, which we have edited for clarity: “In this study, the DCP1.0-M6 sample showed the highest grafting yield and was therefore selected as the compatibilizer for the proposed PLA/PBSeT blends. Hereafter, PLA-g-MAH refers to DCP1.0-M6.” We have also added this information as a footnote to Table 2 (formerly Table 1) to clarify this selection. Although the grafting efficiency may be an important factor for commercial applications, most related research has focused on the grafting yield.
We have measured the mechanical properties of blends using only DCP1.0-M6 (a type of PLA-g-MAH) as a compatibilizer, and we observed that adding 9 phr of DPC1.0-M6 to a PLA/PBSeT blend provided the highest elongation (119.3%).
9. Grafted
(Answer) We have corrected this word on line 181.
10.Recommend original labeling
(Answer) Thank you for your recommendation. We have edited Figure 9 according as you suggested.
Round 2
Reviewer 1 Report
The authors responded to all comments of the reviewer. I recommend publishing the manuscript in IJMS.